

# A new method for estimating UV fluxes at ground level in cloud-free conditions

William Wandji Nyamsi[1,2], Mikko R. A. Pitkänen[2,3], Youva Aoun[1], Philippe Blanc[1], Anu Heikkilä[4], Kaisa Lakkala[4,5], Germar Bernhard[6], Tapani Koskela[7], Anders V. Lindfors[2], Antti Arola[2] and Lucien Wald[1]

[1]MINES ParisTech, PSL Research University, Centre Observation, Impacts, Energy, Sophia Antipolis, France
[2]Finnish Meteorological Institute, Kuopio, Finland
[3]Department of Applied Physics, University of Eastern Finland, Kuopio, Finland
[4]Finnish Meteorological Institute, Climate Research, Helsinki, Finland
[5]Finnish Meteorological Institute, Arctic Research, Sodankylä, Finland
[6]Biospherical Instruments Inc., San Diego, California, USA
[7]Independent researcher, Helsinki, Finland

*Correspondence to*: William Wandji Nyamsi (william.wandji@fmi.fi)

**Abstract.** A new method has been developed to estimate the global and direct solar irradiance in the UV-A and UV-B, at ground level in cloud-free conditions. It is based on a resampling technique applied to the results of the *k*-distribution method and the correlated-*k* approximation

20     of Kato et al. (1999) over the UV band. Its inputs are the aerosol properties, and total column ozone that are produced by the Copernicus Atmosphere Monitoring Service (CAMS). The estimates from this new method have been compared to instantaneous measurements of global UV irradiances made in cloud-free conditions at five stations at high latitudes in various climates. For the total or UV-A global irradiance, the bias ranges between -0.8 W m$^{-2}$ (-3% of

25     the mean of all data) and -0.2 W m$^{-2}$ (-1%). The root mean square error (RMSE) ranges from 1.1 W m$^{-2}$ (6%) to 1.9 W m$^{-2}$ (9%). The coefficient of determination R$^2$ is greater than 0.98. The bias for UV-B is between -0.04 W m$^{-2}$ (-4%) and 0.08 W m$^{-2}$ (+13%) and the RMSE is 0.1 W m$^{-2}$ (between 12% and 18%). R$^2$ ranges between 0.97 and 0.99. This work demonstrates the quality of the proposed method combined with the CAMS products. Improvements, especially

30     in the modelling of the reflectivity of the Earth's surface in the UV region, are necessary prior its inclusion into an operational tool.



## 1. Introduction

Solar UV radiation at the Earth's surface has beneficial and adverse effects on human health (Juzeniene et al., 2011). The wavelength dependence of these effects is typically characterized by action spectra. The most widely used one is the standardized action spectrum for erythema, also known as the CIE (Commission Internationale de l'Eclairage) spectrum (McKinlay and Diffey, 1987). There are also other action spectra related to skin cancer and melanoma (de Gruijl et al., 1993; Setlow et al., 1993). Emphasis has been put mostly on the assessment of the solar UV-erythemal irradiance, also known as the UV-CIE, and the derived quantity, the UV index, a very popular quantity to inform the public about UV levels. The UV index is also used in campaigns promoting healthy sun exposure. While the UV-B (280–320 nm) band is the major contributor to UV-CIE, interest is growing in the role of UV-A (320–400 nm) and total UV (280–400 nm) on various diseases, such as viral infections (Norval, 2006), multiple sclerosis (Orton et al., 2011), eye diseases (Delcourt et al., 2014), skin cancer (Coste et al., 2015; Fortes et al., 2016), or thyroid cancer (Mesrine et al., 2017), among many others (Juzeniene et al., 2011; Norval and Halliday, 2011).

Ground-based spectroradiometers are one of the means to monitor the intensity of solar UV fluxes. Such measurements are rare due to high costs of the instruments, operations and maintenance. To overcome this scarcity, many researchers have looked for proxies and have studied the relationship between UV radiation and the surface downwelling solar radiation integrated over the whole spectrum (280–4000 nm), called total or broadband radiation, since the latter is measured at a greater number of stations or can be estimated at any place from satellite images (Blanc et al., 2011; Lefèvre et al., 2014). Several empirical relationships have been published that relate, with the knowledge of the total ozone column (TOC), the total irradiance to the UV-CIE (den Outer et al., 2010; Calbo et al., 2005) or the UV-A, UV-B or total UV irradiance (Aculinin et al., 2016; Canada et al., 2003; Foyo-Moreno et al, 1998).

An alternative way is the use of an appropriate radiative transfer model (RTM) together with accurate inputs describing the state of the atmosphere in cloud-free condition and the properties of the ground surrounding the instrument. libRadtran is such a RTM (Emde et al., 2016; Mayer and Kylling, 2005). A comparison between 1200 measured UV spectra and estimates made with a previous version called uvspec —now part of libRadtran— with only ozone and aerosol optical properties as inputs yielded very good performance to simulate the UV irradiance under



cloud-free conditions (Mayer et al., 1997). The relative biases ranged between -11% and +2% for wavelengths between 295 nm and 400 nm and solar zenith angles up to 80°. Using measurements from sites in Finland, Norway and Sweden, Lindfors et al. (2007, 2009) showed that UV-CIE and spectral UV irradiances can be accurately modelled using libRadtran and

broadband radiation, TOC, the total water vapor column from the ERA-40 data set, the surface albedo as estimated from snow depth, and the altitude of the location as input.

RTMs are usually computationally expensive; about hundreds of spectral calculations are required to compute the total UV irradiance in a RTM. Strategies have been built to reduce the amount of calculations. Among them, are the *k*-distribution method and correlated-*k*

approximation by Kato et al. (1999). The approach was originally designed for the calculation of the total solar irradiance. It consists in calculating the broadband solar irradiance with only 32 spectral calculations in the spectral range between 240 nm and 4606 nm. The operational McClear model estimating the broadband irradiance in cloud-free conditions accurately reproduces the irradiance computed by libRadtran based on the Kato et al. approach (Lefèvre

et al.; 2013). The McClear model uses several abaci computed by libRadtran for selected values of inputs and provides irradiance at each of 32 spectral intervals. Hereafter, these 32 spectral intervals are named Kato bands and abbreviated KB with the number in subscript. Four KB are covering the whole UV range: $KB_3$ [283, 307] nm, $KB_4$ [307, 328] nm, $KB_5$ [328, 363] nm and $KB_6$ [363, 408] nm. In $KB_1$ and $KB_2$, atmospheric ozone attenuates the radiation before it

reaches the ground.

Wandji Nyamsi et al. (2014) compared atmospheric transmissivities obtained by the Kato et al. approach against those obtained by spectrally resolved computations using two RTMs in each of the 32 KBs. These calculations were performed for a set of 200,000 realistic atmospheres and clouds. As for the UV band, the authors found that the Kato et al. approach offers very

accurate estimates of irradiances in $KB_5$ and $KB_6$. On the contrary, a very large underestimation of the transmissivity was observed in $KB_3$ [283, 307] nm and $KB_4$ [307, 328] nm. This is due to the assumption made by Kato et al. (1999) that in these bands, a single ozone cross section at the central wavelength is sufficient to accurately represent the absorption by ozone over the whole interval. In a subsequent work, Wandji Nyamsi et al. (2015b) have proposed a novel

parameterization using more than one single ozone cross sections which accurately represents the transmissivity due to ozone absorption. Version 2.0.1 of libRadtran (Emde et al., 2016) includes this correction and has been used in this study.





The KBs do not fit exactly the UV spectral ranges and a spectral resampling is necessary. This
is the subject of the present article. The concept of the novel method is to determine several 1-
nm spectral bands whose atmospheric transmissivities are correlated to those of the KB and
then use these transmissivities in a linear interpolation process to compute the UV irradiance.

The method is empirically implemented by the means of libRadtran in cloud-free conditions.
The concept has already been tested for photosynthetically active radiation (PAR) simulated
by libRadtran (Wandji Nyamsi et al., 2015a). Now, the concept is tested for actual UV fluxes.
This work is part of a larger project whose overarching goal is to create an operational tool for
estimating UV fluxes. In particular, it exploits the recent results on aerosol properties, and total

column ozone produced by the Copernicus Atmosphere Monitoring Service (CAMS) for any
location and any time after 2003. The performance of the novel method is assessed by a
comparison against high quality measurements of UV fluxes performed at five stations located
at high latitudes.

**2. Description of measurements used for comparison**

Ground-based measurements were collected from three sites of the UV network of the National
Science Foundation (NSF) of the U.S.A. and two sites of the Finnish Meteorological Institute
(FMI). Table 1 reports the geographical coordinates of the stations, time period of data and
their source, type of instruments, spectral interval and step of measurements.

The site 'Barrow' is located approximately 6 km northeast of the Barrow city in Alaska, on the
coast of the Chukchi Sea, part of the Arctic Ocean, usually covered by ice between November
and July. The snow cover of the surroundings of the station extends from October till June.
According to Bernhard et al. (2008), the effective UV albedo of the surface reaches its
maximum of approximately 0.8 during March-April. It decreases to 0.05 in summer from

August until September.

The site 'Sodankylä' is approximately 6 km southwards the village of Sodankylä. The site is
located in the vicinity of the river Kitinen, and the surroundings are boreal pine forest and large
peatland areas. A permanent snow cover is present during the winter and the annual number of
snowy days is on average 190. The snow cover starts accumulating in October or November

and melts away during May almost every year. The effective UV albedo follows a seasonal





variation due to the snow cover. It ranges from very low values in boreal summer up to 0.65 in winter (Arola et al.; 2003).

Jokioinen Observatory is located in a fairly flat rural area in the southwest of Finland surrounded by fields of cultures and a boreal forest. The number of snowy days is typically 130. The snow conditions in Jokioinen vary from year to year, and also within each winter. At earliest, snow may appear at the end of October or early November, while it typically melts away in March or April. The effective UV albedo is highly variable; it may rise up to 0.58 in boreal winter but more typical values are 0.2 to 0.5 (Lindfors et al., 2007).

Palmer is situated on the Anvers Island, on the western side of the Antarctic Peninsula. The ocean surrounding the island is frozen during austral winter and usually ice free in summer. According to Bernhard et al. (2005), the effective UV albedo varies between 0.6 and 0.95 between August and November, and then decreases down to 0.3 to 0.5 after snowmelt. It is large even in austral summer because of the glaciers surrounding the site.

McMurdo is a coastal site located on the Ross Island, a volcanic island of Antarctica surrounded by a quasi-persistence of the ice sheet. The surroundings of the station are mostly made of dark volcanic rocks. McMurdo has an annual cycle of change in effective UV albedo. It ranges between 0.54 (March) and 0.99 (October) (Bernhard et al., 2006).

WMO (2008) reports that uncertainties associated with the measurements in UV by spectrometers is difficult to estimate precisely. Beyond the technical specifics of the site itself, several errors may occur in the calibration of the instrument that include *i)* the uncertainties associated with irradiance transfer standards, *ii)* the stability of instruments over time, and *iii)* imperfect directional response. The WMO guide estimates that a 5% measurement uncertainty at 300 nm can be achieved only under the most rigorous conditions at the present time.

The data provided by the NSF are available online and can be downloaded freely. Only data of version 2, which have been corrected for the instruments' cosine error, have been selected for ensuring higher accuracy. Data measured during clear sky conditions are flagged (flag "CS"). The number of clear sky instants is reported in Table 1. Integrated irradiances in the UV-A and UV-B range are available and have been downloaded from the website http://uv.biospherical.com/Version2.



The data for the two Finnish sites has been corrected for all known errors following the routine spectral UV data processing procedure of the FMI (Lakkala et al., 2008; Mäkelä et al., 2016). In this work, the measured UV spectra have been first deconvoluted and then convoluted with a standard triangular slit function with a FWHM of 1 nm and extrapolated using the SHICrivm

software package to cover the full UV spectrum. In other words, the software combines measured spectra with an adjusted extraterrestrial solar spectrum to obtain a standardized irradiance, as explained by Tanskanen et al. (2007). It is available and documented at http://tinyurl.com/hdhn6z9. Spectral irradiances are integrated over the UV-B, and UV-A.

In addition, at both Finnish stations, direct, diffuse and global broadband irradiances are
measured every 1 min, the global irradiance being the sum of the direct and diffuse irradiances on a horizontal plane. These series of data are exploited for selecting clear sky instants by using the very restrictive algorithm proposed by Lefèvre et al. (2013). The latter is made of two successive filters. The first one is a constraint on the amount of diffuse irradiance with respect to the global irradiance since the direct irradiance is usually prominent in the case of clear sky.
The second filter analyses the temporal variability of the global irradiance. If there is no cloud, the sky should be clear and steady for a long period.

### 3.  Description of the new method

In brief, the method combines the fluxes estimated by libRadtran in the four KBs and performs
a resampling of these fluxes for retrieving UV fluxes.

### 3.1.      Inputs to libRadtran

In cloud-free conditions, UV irradiance depends mostly on the solar zenith angle $\Theta_s$, the ground albedo, the total column content of ozone, the vertical profile of temperature, pressure, density, and volume mixing ratio for gases as a function of altitude, aerosol optical depth (AOD),
Ångström coefficient, aerosol type and the elevation of the ground above sea level. As the method shall be used operationally, the sources of these inputs have been selected accordingly to allow estimation of UV irradiance at any location and any time.

The Copernicus Atmosphere Monitoring Service (CAMS) of the European Commission is providing aerosol properties together with physically consistent total column ozone for any



place and any time after 2003. Total column content of ozone, as well as the (AOD) at 550 nm, Ångström coefficient, and aerosol type, are collected from this source of data following exactly the path of the McClear model (Lefèvre et al., 2013). $\Theta_s$ is given by the SG2 algorithm for the sun position and angles (Blanc and Wald, 2012). Ground elevation is extracted from one of

several available databases, such as the SRTM (Shuttle Radar Topography Mission).

The albedo is the ratio of upwelling to downwelling flux and is the integral of the bidirectional reflectance distribution function (BRDF), which depends on the surface-type, its roughness, and the wavelength of the impinging radiation. A few institutes provide BRDF products in the UV range or Lambert equivalent reflectances (e.g., Herman and Celarier, 1997) covering the

world with coarse resolution of 0.5 or 1°. In absence of the ideal solution —BRDF parameters in the UV range available worldwide with a grid cell of 0.05° or better— an approximate solution has been adopted by using the broadband BRDF parameters proposed by Blanc et al. (2014). The U.S. National Aeronautics and Space Administration (NASA) provides worldwide maps of the BRDF parameters that are derived from the MODIS (Moderate Resolution Imaging

Spectroradiometer) instrument (Schaaf et al., 2002). Blanc et al. (2014) have created a series of maps of the MODIS BRDF parameters for each calendar month for the broadband albedo with no missing values at a spatial resolution of 0.05°. In addition, these authors proposed a method for computing the albedo simultaneously for direct and global irradiances. These maps and this method are those used by the McClear model (Lefèvre et al., 2013). This assumption

may be crude depending of the surface. For example, Varotsos et al. (2014) reported from many aircraft measurements that albedo is typically less than 0.08 in the UV for most surfaces but that the albedo changes dramatically in the NIR depending on the surface, with values as high as 0.8 for sand and as low as 0.04 for water. As for the snow, its state is of crucial importance for the profile of its spectral albedo dependence which exhibits a tendency to decrease from

UV to the NIR independently of the sky conditions.

### 3.2.    Resampling technique

Let $\lambda$ be the wavelength, $G_\lambda$ the global spectral irradiance at the surface and $B_\lambda$ the direct normal spectral irradiance, i.e. the irradiance received at the surface on a plane normal to the sun's rays from the direction of the sun. The irradiance in a given interval $[\lambda_1, \lambda_2]$ is given by:

$$G_{\lambda 1 \lambda 2} = \int_{\lambda_1}^{\lambda_2} G_\lambda \, d\lambda \qquad (1)$$





Similar expressions may be obtained for irradiances in UV-A, UV-B, total UV or over $KB_j$. For example, the total UV irradiance $G_{UV}$ and the direct normal irradiance $B_{UV}$ or the irradiance $G_{KB4}$ in $KB_4$ are given by:

$$G_{UV} = \int_{280}^{400} G_\lambda \, d\lambda \qquad (2)$$

$\quad B_{UV} = \int_{280}^{400} B_\lambda \, d\lambda \qquad (3)$

$$G_{KB4} = \int_{307}^{328} G_\lambda \, d\lambda \qquad (4)$$

where $\lambda$ is expressed in nm.

The KBs do not fit exactly the UV spectral ranges. For example, the UV-B is covered by $KB_3$ and a part of $KB_4$. One solution to estimate the irradiance in a UV interval is the use of weighted

sums based on the overlap between $KB_j$ and this interval. Another technique is adopted here whose concept is to determine several 1-nm spectral bands $NB_i$ whose transmissivities are correlated to those of the $KB_j$ and then use these transmissivities in a linear interpolation process to compute the UV irradiance. A similar approach has been used by Wandji Nyamsi et al. (2015a) for the calculation of the photosynthetically active radiation with better results than

a weighted sum.

If one assumes that the optical properties of the atmosphere do not change over a given $NB_i$, the integrals of Eqs. (2) – (4) may be replaced by Riemann sums over $NB_i$. For example, if $\lambda_i$ denotes the central wavelength of $NB_i$, $G_{UV}$ can be approximated by:

$$G_{UV} = \sum_{i=1}^{120} G_{\lambda i} \qquad (5)$$

$\quad$ If one defines the clearness index $KT_i$ as:

$$KT_i = \frac{G_{\lambda i}}{Eo_i \cos(\theta_s)} \qquad (6)$$

where $Eo_i$ is the irradiance at the top of atmosphere on a plane normal to the sun's rays for $NB_i$ for a given instant $t$, Eq. (5) becomes

$$G_{UV} = \cos(\theta_s) \sum_{i=1}^{120} Eo_i \, KT_i \qquad (7)$$



Similarly, the clearness index $KT_{KBj}$ for $KB_j$ is given by:

$$KT_{KBj} = \frac{G_{KBj}}{Eo_{KBj} \cos(\theta_s)} \qquad (8)$$

The solar spectrum of Gueymard (2004) is available in libRadtran. Together with the algorithm SG2 for computing the sun position (Blanc and Wald, 2012), it has been used here to compute
$Eo_i$ and $Eo_{KBj}$.

For the method presented here, we assume that simple and accurate relations, e.g. affine functions, may be found between each $KT_{KBj}$ and a sub-set of several $KT_i$, called $KT_k$ hereafter. Then it may be possible to interpolate linearly between $KT_k$ to obtain an estimate of $KT_i$ for each $\lambda_i$. By summing the products of $Eo_i$ and these interpolated $KT_i$ over a given spectral
interval, it is then possible to compute the corresponding irradiance. This is the principle of the resampling technique. The set of $NB_k$ is selected at the beginning of the procedure and the same set is used for all processing. The number of $NB_k$ should be as small as possible in order to decrease the amount of calculations but still large enough to allow a good accuracy.

The selection of $NB_k$ is empirically determined by means of libRadtran. The current approach
is empirical with no guarantee that the selected set of $NB_k$ is the optimum. It could have been possible to use some mathematical optimisation tools. This is not a straightforward process as the cost function should take into account that the number of $NB_k$ is unknown *a priori*.

A set of 60 000 clear sky atmospheric states was built by means of the Monte-Carlo technique in order to select $NB_k$. Each state comprises the nine variables described above, and the value
of each variable was randomly selected by taking into account their modelled marginal distribution. The distributions proposed by Lefèvre et al. (2013) and Oumbe et al. (2011) established from observations were adopted here (Table 2). More specially, the uniform distribution is chosen as a model for marginal probability for all variables except AOD, Angstrom coefficient, and total column content of ozone. The chi-square law for AOD, the
normal law for the Angstrom coefficient, and the beta law for total column content of ozone have been selected. The selection of these parametric probability density functions and their corresponding parameters have been empirically determined from the analyses of the observations made in the AERONET network for aerosol properties and from meteorological satellite-based ozone products (Lefèvre et al., 2013).





Each atmospheric state is input twice to libRadtran, (1) with the Kato et al. approach yielding $KT_{KB3}$, $KT_{KB4}$, $KT_{KB5}$ and $KT_{KB6}$, and (2) with detailed spectral computations providing $KT_i$ every 1 nm for the interval [283, 408] nm. Several plots were made superimposing $KT_{KB3}$, $KT_{KB4}$, $KT_{KB5}$ and $KT_{KB6}$ (in green) and $KT_i$ (in red). Figure 1 is such a graph with the following

inputs: $\Theta_s$ of 61.33°, *midlatitude summer* atmospheric profile, TOC of 286 DU, AOD of 0.274 at 1000 nm for a urban aerosol model with an Ångstrom exponent of 0.87, elevation of 1000 m and surface albedo of 0.83. A visual inspection shows that $KT_{KBj}$ and $KT_i$ are approximately equal for $\lambda_j$ in the middle of $KB_j$, excepted for $KB_5$. $KT_i$ in this band exhibits a non-linear behaviour that cannot be accounted for with a single $KT_k$. If one selects 305, 320, 333, 346 and

386 nm as $NB_k$ (magenta crosses), then the linear interpolation (in blue) provides a fairly accurate estimate of $KT_i$. The five $NB_k$ were selected by a lengthy visual inspection of such plots and are reported in Table 3. The same $NB_k$ apply for the global and direct irradiances. For each $NB_k$, the parameters of the affine function relating $KT_{KBj}$ and $KT_k$ are determined by least-square fitting technique (Table 3):

$$KT_k = a_k\, KT_{KBj} + b_k \qquad\qquad (9)$$

Another set of parameters is determined in the same way for the direct irradiance. In the operational mode, given an atmospheric state, a run of libRadtran, or a fast approximation of it, yields four $KT_{KBj}$, from which the five $KT_k$ are computed using the affine functions. Then, approximate $KT_i^*$ are computed for each nm between 280 and 400 nm using a linear

interpolation and extrapolation of $KT_k$. In case extrapolation provides negative values, $KT_i^*$ is set to 0. Eventually, $G_{UV}$ is obtained by

$$G_{UV} = cos(\Theta_s) \sum_{i=1}^{120} Eo_i\ KT_i^* \qquad\qquad (10)$$

A similar process is performed for the direct normal irradiance $B_{UV}$ as well as for the same quantities in UV-A and UV-B. As the method provides the spectrum $KT_i^*$, the equation may

be extended to include any action spectrum $S(\lambda)$, e.g.:

$$G_{S(\lambda)} = cos(\Theta_s) \sum_{\lambda 1}^{\lambda 2} Eo_i\ S(i)\ KT_i^* \qquad\qquad (11)$$





### 3.3.  Numerical validation

In this section, results of the proposed technique are compared with results from the detailed spectral calculations made by libRadtran to assess the accuracy of the proposed technique for $G_{UVA}$, $G_{UVB}$, $B_{UVA}$ and $B_{UVB}$ for cloud-free conditions. The errors made by using the proposed

technique for calculations of the UV-A and UV-B irradiances are presented. To that extent, an additional sample of 10 000 atmospheric states has been randomly constructed following the marginal distribution variables described in Table 2. The proposed technique was applied to the outputs of libRadtran using Kato et al. approach and the estimates were compared to the detailed calculations performed by libRadtran. Following the ISO standard (1995), the

deviations were computed by subtracting measurements for each instant from the results of the method. They were summarized by the bias (mean error), the root mean square error (RMSE), and their values *rbias* and *rRMSE* relative to the mean value of the measurements. In addition, the coefficient of determination ($R^2$) is computed.

Table 4 reports the statistical indicators for the global and direct normal UV-A, UV-B

irradiances. For UV–A fluxes, the bias for the global irradiance and the direct irradiance, is +0.10 W m$^{-2}$, i.e. +0.23% in relative value, and -0.15 W m$^{-2}$, i.e. -0.65% in relative value, respectively. The RMSE is respectively 0.12 W m$^{-2}$ (0.25%) and 0.18 W m$^{-2}$ (0.78%). For UV–B fluxes, the bias for the global irradiance and the direct irradiance, is -0.04 W m$^{-2}$, i.e. -1.64% in relative value, and +0.07 W m$^{-2}$, i.e. +10.10% in relative value, respectively. The

corresponding RMSE is respectively 0.14 W m$^{-2}$ (6.19%) and 0.15 W m$^{-2}$ (20.48%). The coefficient of determination $R^2$ is greater than 0.99 except for the direct normal UV-B irradiance which is 0.966. Expectedly, these indicators prove the good level of performance of the proposed technique.

### 4. Results

The results of the proposed method were compared to measurements of UV–A and UV–B irradiances at the surface for cloud-free conditions. Similar statistical indicators as those presented in the previous section are also computed to synthetize the errors.


### 4.1. Performance of the method for UV–A irradiance

Figure 2 exhibits the 2D histogram between ground-based instantaneous measurements made at Barrow for cloud-free conditions and estimates from the proposed method combined with inputs from CAMS. All points are well located along the identity line. The slope of the fitting line is 0.995, i.e. very close to 1, showing a very good estimation of the measurements by the method. $R^2$ is 0.97, meaning that all the variability in the measurements is very well explained by the estimates. The bias is low: -0.2 W m$^{-2}$, i.e. -1% of the mean value of the measurements: 20.5 W m$^{-2}$. The RMSE is small: 1.4 W m$^{-2}$, around 7% of the mean. These statistical indicators for each station are reported in Table 5. The measurements are mostly between March and September. The broadband albedo is 0.8 from the beginning of March till the middle of May. With the progressive snowmelt, this broadband albedo decreases from mid-May down to 0.12 at mid-July. This variation corresponds well to the climatological evolution reported by Bernhard et al. (2008) and supports the choice of this approximation by the broadband albedo.

Results for Sodankylä are shown in Figure 3. Cloud-free conditions occur mostly between February and September. The points lie along the identity line with a slight overestimation by the method at low irradiance and an underestimation at large irradiance. $R^2$ is 0.98. The bias is low: -0.5 W m$^{-2}$, i.e. -2% of the mean value: 20.8 W m$^{-2}$. The RMSE is low: 1.9 W m$^{-2}$ (9%). As for Jokioinen (Figure 4), all points are well located along the identity line. $R^2$ is 0.98. The bias is low: -0.5 W m$^{-2}$, i.e. -2% of the mean value of the measurements: 22.1 W m$^{-2}$, as well as the RMSE: 1.6 W m$^{-2}$ (8%).

Results for Palmer are shown in Figure 5. One may note that the points are well aligned with low scatter along a straight line whose slope is 0.99 with a slight underestimation by the method. The bias is -0.8 W m$^{-2}$ (-3% of the mean value of 24.9 W m$^{-2}$). The RMSE is 1.2 W m$^{-2}$ (5%). $R^2$ is greater than 0.99. Cloud-free conditions occur mostly between August and April. The broadband albedo slightly increases from 0.28 to 0.32 between August and March, then decreases until April up to 0.20. These values are small and close to those of a ground free of snow or ice. The effective UV albedo is usually greater than 0.3 with peaks up to 0.8. As a smaller albedo means a smaller contribution to the diffuse part of the irradiance, the difference between the broadband and effective UV albedo may explain the slight underestimation indicated in Figure 5.





Results for McMurdo are shown in Figure 6. Cloud-free conditions occur mostly between April and September. The points are aligned along the identity line. $R^2$ is 0.99. The bias is low: -0.3 Wm$^{-2}$ (-1% of the mean value: 21.0 W m$^{-2}$), as well as the RMSE: 1.2 W m$^{-2}$ (6%). The broadband albedo may reach 0.8 and there is no clear discrepancy between the broadband and effective UV albedo. Nevertheless, the authors believe that the outliers may be explained by a difference in effective UV albedo.

The dependence of errors by solar zenith angle was investigated. Figure 7 exhibits the statistical indicators as a function of $\Theta_s$ for each station: bias and RMSE on the left side and rbias and rRMSE on the right side. The absolute values of the bias show a tendency to decrease as $\Theta_s$ increases, with the maximum reached for low $\Theta_s$, except McMurdo. In the opposite manner, the absolute values of the relative bias show a tendency to increase with $\Theta_s$. This could be related to the fact that low $\Theta_s$ are reached in summer with greater values in UV irradiance and lower values in effective UV albedo.

## 4.2. Performance of the method for UV-B irradiance

The UV-B band is the spectral region of UV irradiance where the ozone absorption is very strong. Figure 8 exhibits the 2D histogram between ground-based instantaneous measurements made for each station in cloud-free conditions and estimates from the proposed method. The station name is indicated on the plot of each histogram. Table 6 reports the statistical indicators for UV-B. $R^2$ is greater than 0.97 for all stations, meaning that variability in UV-B is very well reproduced by the estimates. In general, the method overestimates the UV-B irradiances. Visually, one observes that the method clearly overestimates when the irradiance is low. Further investigation reveals a systematic overestimation at the low irradiance from the method itself. This mainly explains this previous observation. The absolute value of the bias is less than 0.1 W m$^{-2}$. The relative bias ranges between –4% (Palmer) and 13% (Barrow). rRMSE ranges between 12% and 18%.

Figure 9 shows the change in bias, rbias, RMSE and rRMSE as a function of $\Theta_s$. One may observe a tendency of the bias to reach a maximum between 65° and 75°; this appears in the form of a plateau around 65° for the RMSE which decreases as $\Theta_s$ increases, i.e. as the irradiance decreases. The relative bias increases with $\Theta_s$ as well as the rRMSE. Both are closer in the high $\Theta_s$ than in the small $\Theta_s$.





## 5. Discussion and Conclusion

The comparison has demonstrated a reasonable agreement between the ground-based measurements of UV-A and UV-B and the estimates by the proposed method with CAMS products as inputs. The variability in UV fluxes is well reproduced by the method. A good level

of accuracy is reached that is close to the uncertainty of the measurements themselves. The computations of the fluxes in the KBs can be performed quite fast with the use of pre-computed abaci as shown by the example of the McClear model. This model is an accurate approximation of libRadtran but $10^5$ times faster. The proposed method extends the results of the McClear model to the UV range and can be used in future operational tools that are both accurate and

fast.

Further improvements are needed. A major improvement would be the extension to all sky conditions. In this aspect, one may build on the work of Oumbe et al. (2014) who demonstrated that, in the case of an infinite plane-parallel single- and double-layered cloud, the solar irradiance at ground level computed by a radiative transfer model can be approximated by the

product of the irradiance under clear atmosphere and a modification factor that depends on cloud properties and ground albedo only. Changes in clear-atmosphere properties have negligible effect on the latter so that both terms can be calculated independently. Such an approximation has been exploited previously with limited justification by several authors in studies on total irradiance (Huang et al., 2011), UV or photosynthetically active radiation (see

e.g. Calbo et al., 2005; den Outer et al., 2010; Krotkov et al., 2001).

Another improvement consists in the modelling of the surface albedo in the UV range. Maps of BRDF parameters in the UV range must be created with a satisfactory spatial resolution of 0.05° or better. The MODIS BRDF parameters may be a starting point as they are available at several wavelengths. It could be possible to apply the technique used by Blanc et al. (2014) to

create BRDF maps for each wavelength and for each calendar month with no missing values. The smallest wavelength in the MODIS BRDF is approximately 470 nm, i.e. outside the UV range, and extrapolation towards small wavelengths will be necessary.



## 6. Data availability

UV data from Barrow, Palmer Station and McMurdo Station were provided by the NSF UV Monitoring Network, operated by Biospherical Instruments Inc. and funded by the U.S. National Science Foundation's Office of Polar Programs. Version 2 data used here are available
from http://uv.biospherical.com/Version2/Version2.asp

FMI's spectral Brewer UV measurements are available through the European UV Data Base: http://uv.fmi.fi/uvdb/

Products from CAMS can be downloaded from the web site: http://atmosphere.copernicus.eu/

The BRDF maps by Blanc et al. (2014) may be downloaded from the web site:
http://tinyurl.com/jhnzxl5.

*Acknowledgments*. William Wandji Nyamsi was partly supported by Foundation Mines ParisTech.

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



**Table 1**: Description of stations used for validation, ordered by decreasing latitude

| Station | Barrow | Sodankylä | Jokionen | Palmer | McMurdo |
|---|---|---|---|---|---|
| Source | NSF | FMI | FMI | NSF | NSF |
| Latitude (°) | 71.32 | 67.37 | 60.82 | -64.77 | -77.83 |
| Longitude (°) | -156.68 | 26.63 | 23.50 | -64.05 | 166.67 |
| Altitude (m) | 8 | 179 | 104 | 21 | 183 |
| Instrument | SUV-100 spectro radiometer | Brewer spectro photometer MK-II#037 | Brewer spectro photometer MK-III #107 | SUV-100 spectro radiometer | SUV-100 spectro radiometer |
| Average acquisition frequency | 4 spectra per hour | Between 1 and 2 spectra per hour | Between 1 and 2 spectra per hour | 4 spectra per hour | 4 spectra per hour |
| Spectral range (nm) | 280-600 | 290-325 | 286.5-365 | 280-600 | 280-600 |
| Step (nm) | 0.2 in UV-B to 1.0 in visible | 0.5 | 0.5 | 0.2 in UV-B to 1.0 in visible | 0.2 in UV-B to 1.0 in visible |
| Period | 2005-01 to 2010-11 | 2007-01 to 2011-12 | 2007-01 to 2008-12 | 2005-01 to 2010-09 | 2005-01 to 2010-02 |
| NCFI* | 4293 | 2590 | 1140 | 1736 | 10175 |

*NCFI: Number of cloud-free instants.





Table 2. Ranges and statistical distributions of values taken by the solar zenith angle, the ground albedo and the 7 variables describing the clear atmosphere

| Variable | Value |
|---|---|
| Solar zenith angle $\Theta s$ | Uniform between 0 and 89 (degree) |
| Ground albedo $\rho g$ | Uniform between 0 and 0.9 |
| Elevation of the ground above mean sea level | Equiprobable in the set: {0, 1, 2, 3} (km) |
| Total column content of ozone | Ozone content is: $300*\beta + 200$, in Dobson unit. Beta distribution, with A parameter = 2, and B parameter = 2, to compute $\beta$ |
| Atmospheric profiles (Air Force Geophysics Laboratory standards) | Equiprobable in the set {"Midlatitude Summer", "Midlatitude Winter", "Subarctic Summer", "Subarctic Winter", "Tropical", "US. Standard"} |
| Aerosol optical depth at 550 nm | Gamma distribution, with shape parameter = 2, and scale parameter = 0.13 |
| Angstrom coefficient | Normal distribution, with mean=1.3 and standard-deviation=0.5 |
| Aerosol type | Equiprobable in the set {"urban", "rural", "maritime", "tropospheric", "desert", "continental", "Antarctic"} |





Table 3. KB covering UV band and selected sub-intervals $NB_k$, slopes and intercepts of the affine functions between the clearness indices in KB and sub-intervals $NB_k$.

| KB | KB range, nm | Sub-interval $NB_k$, nm (# k) | Global | | Direct normal | |
|---|---|---|---|---|---|---|
| | | | Slope $a_i$ | Intercept $b_i$ | Slope $c_i$ | Intercept $d_i$ |
| 3 | 283 – 307 | 304 – 305 (#1) | 3.0900 | 0.0007 | 3.0852 | 0.0003 |
| 4 | 307 – 328 | 319 – 320 (#2) | 1.1264 | -0.0175 | 1.0886 | -0.0007 |
| 5 | 328 – 363 | 332 – 333 (#3) | 1.0247 | -0.0519 | 0.8992 | -0.0103 |
| | | 345 – 346 (#4) | 0.9946 | 0.0152 | 1.0112 | -0.0004 |
| 6 | 363 – 408 | 385 – 386 (#5) | 1.0030 | -0.0032 | 0.9987 | -0.0023 |





Table 4. Statistical indicators of the performances of the proposed technique for estimating UV fluxes

| UV fluxes | Mean (W m$^{-2}$) | Bias (W m$^{-2}$) | RMSE (W m$^{-2}$) | rBias (%) | rRMSE (%) | $R^2$ |
|---|---|---|---|---|---|---|
| $G_{UVA}$ | 45.6 | +0.1 | 0.1 | +0.2 | 0.2 | 1.00 |
| $B_{UVA}$ | 23.4 | -0.1 | 0.2 | -0.6 | 0.8 | 1.00 |
| $G_{UVB}$ | 2.30 | -0.04 | 0.14 | -1.64 | 6.19 | 1.00 |
| $B_{UVB}$ | 0.73 | +0.07 | 0.15 | +10.10 | 20.48 | 0.97 |



Table 5. Statistical indicators of the performances of the method for UV-A irradiance

| Station | Mean (W m$^{-2}$) | Bias (W m$^{-2}$) | RMSE (W m$^{-2}$) | rBias (%) | rRMSE (%) | $R^2$ |
|---------|------|------|------|-------|-------|-------|
| Barrow | 20.0 | -0.2 | 1.4 | -1.1 | 6.8 | 0.98 |
| Sodankylä | 20.8 | -0.5 | 1.9 | -2.5 | 9.0 | 0.98 |
| Jokioinen | 22.1 | -0.5 | 1.6 | -2.1 | 7.5 | 0.98 |
| Palmer | 24.9 | -0.8 | 1.2 | -3.1 | 4.9 | 0.99 |
| McMurdo | 20.3 | -0.3 | 1.1 | -1.6 | 5.6 | 0.99 |



Table 6. Statistical indicators of the performances of the method for UV-B irradiance

| Station | Mean (W m⁻²) | Bias (W m⁻²) | RMSE (W m⁻²) | rBias (%) | rRMSE (%) | $R^2$ |
|---|---|---|---|---|---|---|
| | (W m$^{-2}$) | (W m$^{-2}$) | (W m$^{-2}$) | (%) | (%) | |
| Barrow | 0.57 | 0.08 | 0.10 | 13.41 | 18.01 | 0.97 |
| Sodankylä | 0.65 | 0.05 | 0.09 | 7.74 | 13.91 | 0.98 |
| Jokioinen | 0.75 | 0.05 | 0.10 | 6.70 | 13.74 | 0.98 |
| Palmer | 1.03 | −0.04 | 0.12 | −4.24 | 11.67 | 0.99 |
| McMurdo | 0.72 | 0.04 | 0.09 | 4.86 | 12.32 | 0.98 |



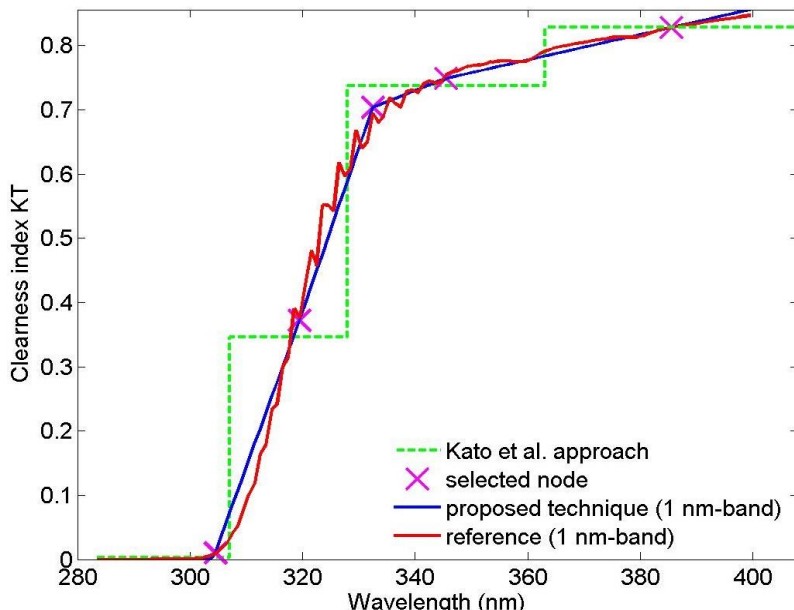

Figure 1: Illustration of the resampling technique



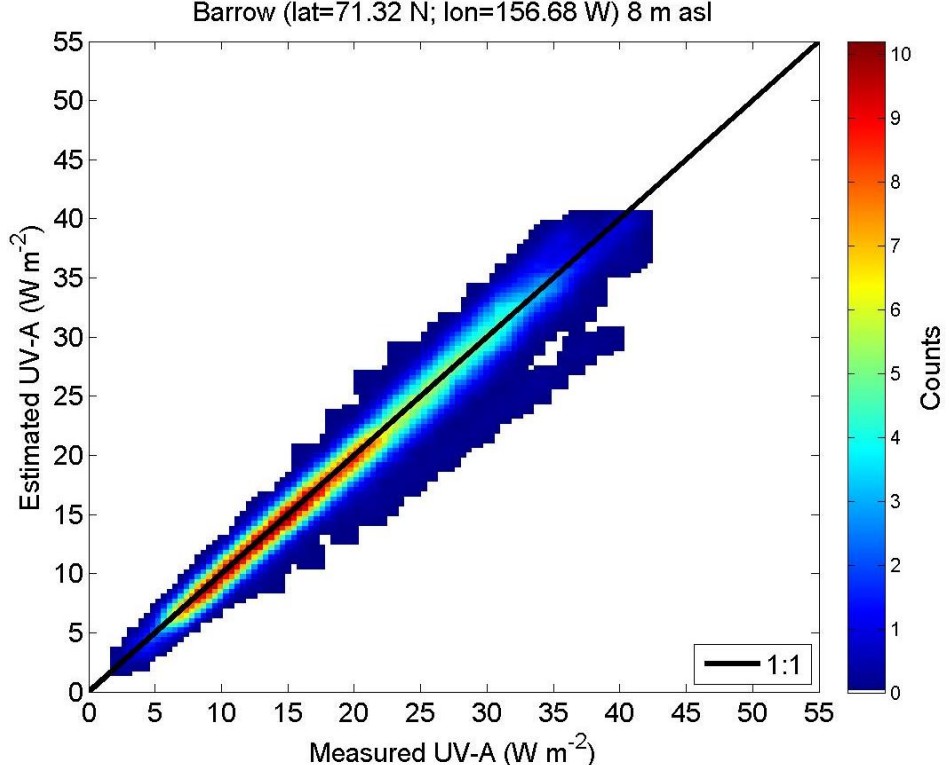

Figure 2: 2D histogram between measurements of UV–A and estimates at Barrow.




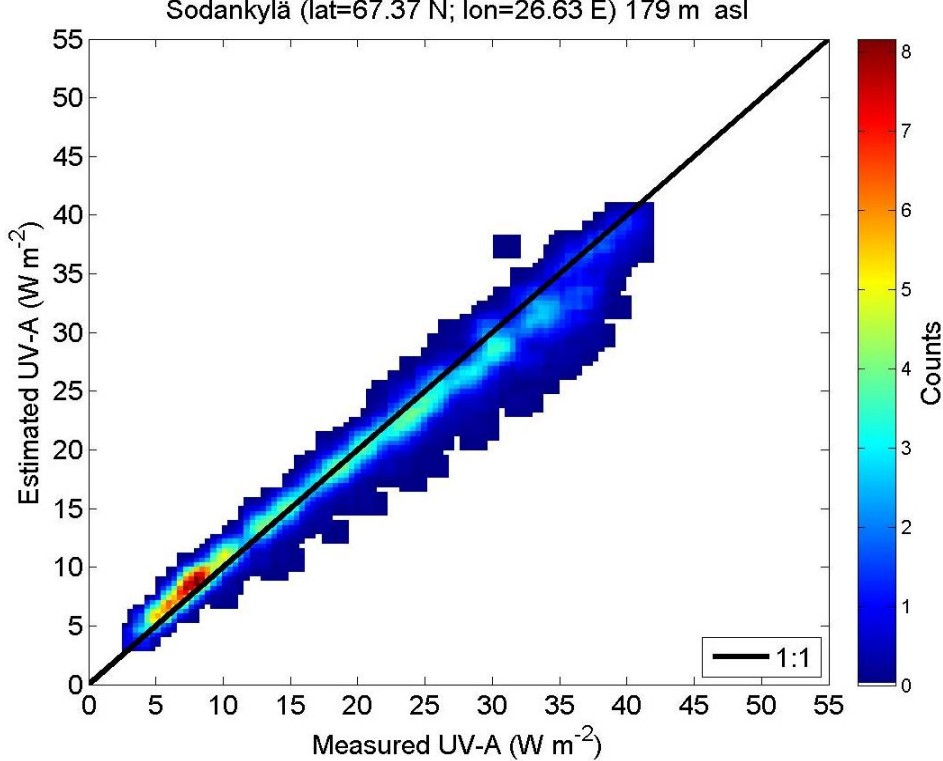

Figure 3: 2D histogram between measurements of UV–A and estimates at Sodankylä

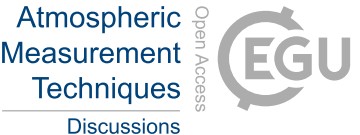

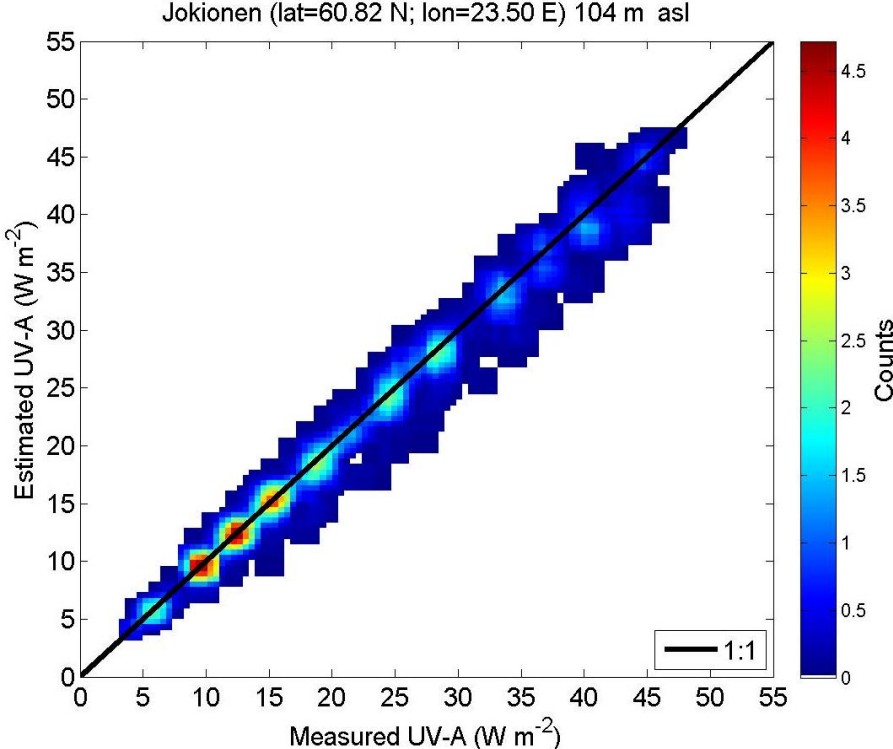

Figure 4: 2D histogram between measurements of UV−A and estimates at Jokioinen





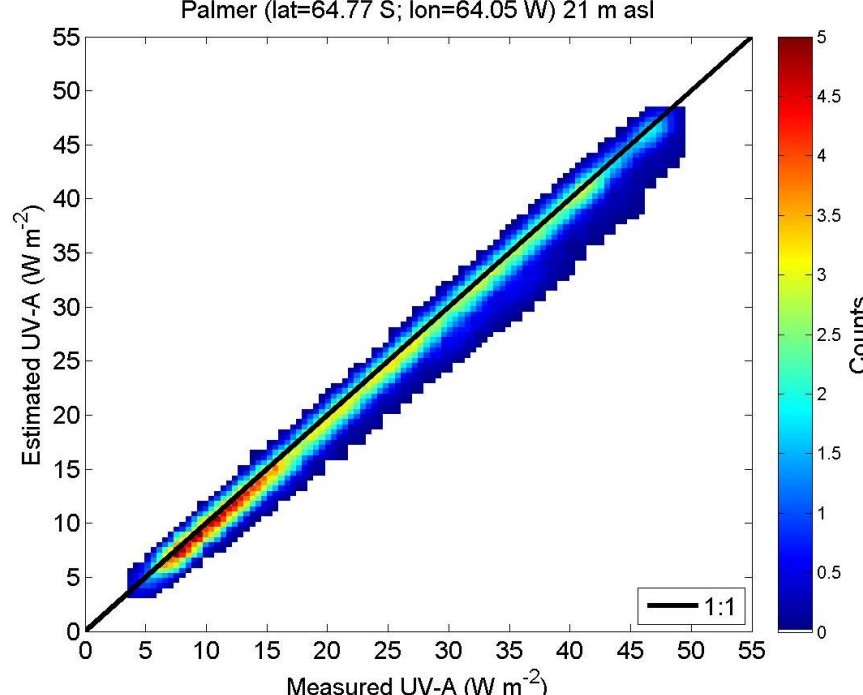

Figure 5: 2D histogram between measurements of UV–A and estimates at Palmer





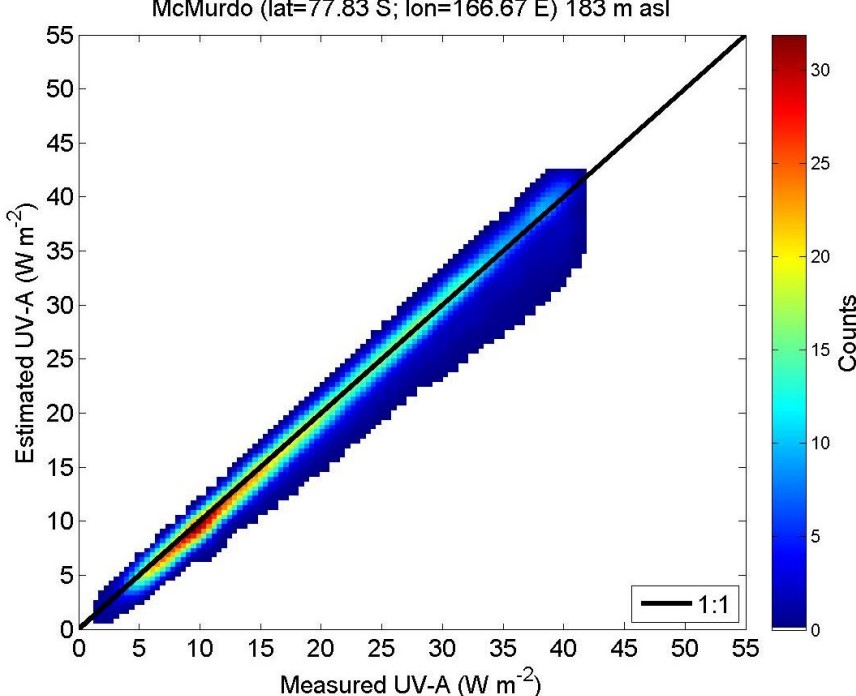

Figure 6: 2D histogram between measurements of UV−A and estimates at McMurdo

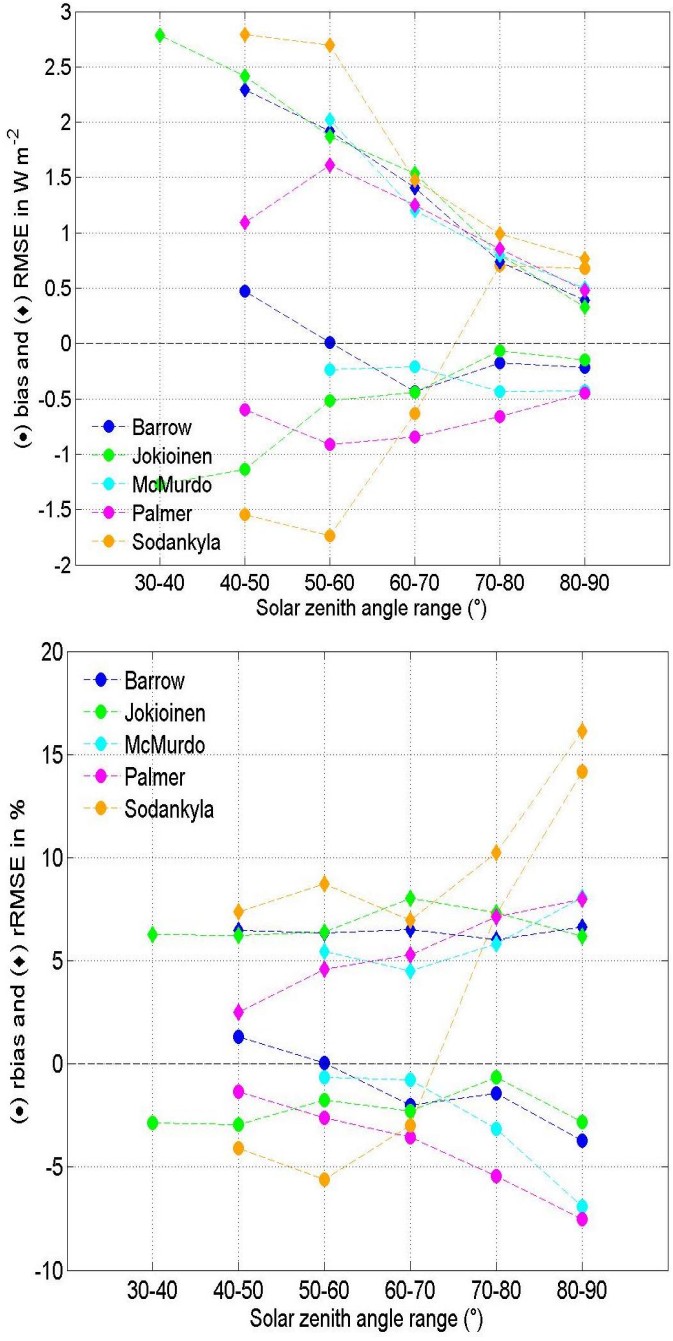

Figure 7: Dependence of bias, RMSE (on the top) and rbias, rRMSE (on the bottom) by solar zenith angle range for total UV–A irradiances for each station





Figure 8: 2D histogram between measurements of UV−B and estimates for each station with each station name on the top.





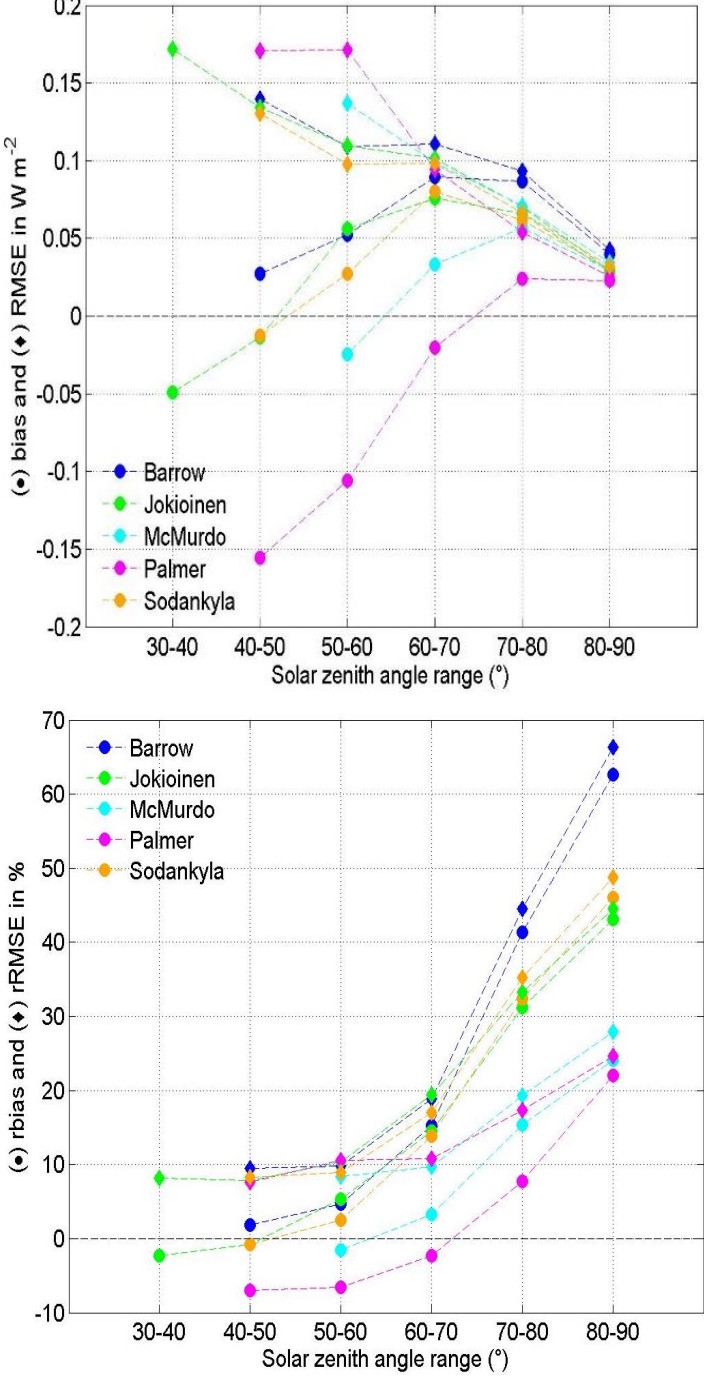

Figure 9: Dependence of bias and RMSE by solar zenith angle for UV-B irradiance for each station