# Peer review of "A new method for estimating UV fluxes at ground level in cloud-free conditions"

_Atmospheric Measurement Techniques, 2017_

## Referee Comment (RC1) · Anonymous Referee #3 · 24 Aug 2017

The manuscript describes a fast method based on a modified k-distribution method and a resampling technique, to estimate ground level UV fluxes. The manuscript is well-organised and include description and verification of new/adopted model results.

The manuscript is acceptable for publication after consideration of the suggestions for changes given below.

- **Page 1, lines 29-31**: It is claimed that improvements in the modelling of the Earth's reflectivity in the UV region are necessary. However, I can not see that the manuscript identifies high reflectivity to be a problem when discussing the model versus measurement results in Figs. 2-6 and 8 or elsewhere in the manuscript. Some weak hints are given, but no strong evidence supports the claim in the

abstract. It should be possible to identify measurements made during high and low albedo situations and compare these with the model to quantify differences caused by differences in the Earth's reflectivity.

- **Page 3, line 25**: "A very large underestimation" is mentioned. To make the manuscript complete and self-contained: may you please include numbers quantifying this underestimate?

- **Page 3, lines 29-30**: Please include numbers quantifying how much the "novel parameterization" improved the transmissivity.

- **Page 6, lines 4-5**: If I understand this correctly, the SHICrivm software is used to obtain the full UV spectrum from the measurements. The Sodankylä Brewer stops at 325 nm while the Jokionen Brewer stops at 365 nm. Thus, for the Sodankylä Brewer nearly the whole UV-A is extrapolated from the UV-B using SHI-Crivm. This approach should be justified and the errors in the extrapolated part assessed and documented. For the Jokionen Brewer the extrapolation is less severe, but needs nevertheless to be justified and the uncertainty discussed.

- **Page 6, line 8**: Please do not use tinyurl and elsewhere. It is just obfuscating.

- **Page 6, lines 15-16**: The sentence "If there is no cloud, the sky should be clear and steady for a long period" is pretty obvious and may be omitted. But maybe you intended to say something else?

- **Page 7, line 5**: Please be a little more specific than "one of several available databases" and mention which one you used, including references and/or URLs.

- **Page 7, lines 20-25**: You seem to be concerned that the albedo changes when approaching the NIR. But that should be of no relevance for the work presented here as you only discuss UV-B and UV-A. Hence, the discussion about NIR albedo may be omitted.
- **Page 9, line 22**: Nine variables are used to build the atmospheric states. One is the solar zenith angle which is sampled uniformly between 0 and 89°. It should be noted that the solar zenith angle dependence of the radiative transfer equation is best described by the cosine of the solar zenith angle and not the solar zenith angle. Hence, in your table 2 it would have been preferable to have $\cos\theta_s$ instead of the $\theta_s$.

- **Page 10, line 8**: What is meant by "excepted for $KB_5$"?

- **Page 13, lines 22-23**: I do not comprehend the sentence "Further investigation reveals a systematic overestimation at the low irradiance from the method itself". What is this systematic overestimation of the method itself? Is there a problem with the method? Why has not that problem been corrected?

- **Page 22, Table 2**: The solar zenith angle is sampled uniformly between 0 and 89°. Was your radiative transfer calculations done in plane-parallel or pseudo-spherical geometry? Please include this information in the model description part.

- **Page 22, Table 2**: Include a column that for each variable gives the total number of samples for each variable (for Aerosol type that is obviously 7, for many of the others it is not possible to tell from the table as is). Also, where applicable, include steps. That is, for uniform distributions you include start and stop, but should also include step size.

- **Page 24, Table 4**: Why is the rBias so much worse for the direct than the global irradiance? Is this due to a worse sampling as in Fig. 1 for the direct irradiance? The global irradiance includes the direct irradiance. Is thus the error in the global irradiance mostly due to the error in the direct irradiance?

- **Pages 25-26, Tables 5-6**: Please include the number of data points included in the analysis for each station. This is valuable information to be able to better

assess the numbers in the tables, as a station with more data points may be considered more "valuable" than one with fewer.

- **Page 27, Fig. 1**: Please indicate (label) where the various $KT_{KB3}$, $KT_{KB4}$, $KT_{KB5}$, and $KT_{KB6}$ bands are on the green line.

- **Pages 27-32, Figs. 2-6**: Please combine these Figures into one as you have already done in Figure 8. Figure 8 is much easier to read and allows for much easier comparison of results from the different stations than Figs. 2-6.

---

## Referee Comment (RC2) · Anonymous Referee #1 · 28 Aug 2017

The manuscript by Wandji Nyamsi et al describes a new, time-efficient method for modeling UV irradiance. The manuscript is concise and well-written, and will be proper for publication in AMT after taking into account the following comments:

In the introduction, few lines of text regarding the reasons for which the authors choose to perform the evaluation of the method using ground based measurements only from high latitude stations (and not from mid-latitudes or the tropics) would be useful.

I suggest using abbreviations for phrases that are often referred in the manuscript (e.g. total column content of ozone could be written as TOC).

I suggest including Figures 2-6 in a single figure, similar to figure 8.

In Figures 2-6 the distribution of the data points around the y=x line is uneven. You

claim that part of this uneven distribution is explained by the imperfect description of the effective UV albedo in the model. In some of the graphs (e.g. in figure 2) there seems to be a "branch" of data where the UV-A is importantly underestimated (∼20% or more) by the model, even for high values of the UV-A (which possibly do not correspond to low SZAs). This branch becomes clearer in the case of UV-B (in all graphs of figure 8). Could you be more specific on what is the cause of this branch (explain more accurately what its cause is, or even provide a graph which proves that this branch is for high/low values of a specific parameter)?
* * *

---

## Referee Comment (RC3) · Anonymous Referee #2 · 14 Sep 2017

The manuscript presents a new method for estimating UV fluxes at ground level in cloud free conditions. The scientific contribution is valuable, the novelty is evident, the organization and structure of the manuscript is good. The manuscript could be considered for publication if the following comments would be taken into account. Major comments 1. Page 6, lines 9-16: the proposed method (Lefevre et al., 2013) for the definition of clear skies is applied on broadband or total irradiance. Is this valid for UV radiation as well? UV radiation is affected considerably more by scattered cloudiness. In this case, you may have an unobstructed Sun (no clouds to cover) and a non-significant effect on diffuse broadband irradiance, so, you can assume that you have a cloud free instant. In UV (direct and diffuse irradiances) however, the effect of scattered cloudiness will be more evident. This is one of the cases that the cloud

modification factors in UV and broadband irradiance are not related with a linear fit. Can you provide some evidence that the propose method is valid for UV as well? 2. Page 7, lines 15-25: it is not clear in the document the type of albedo used as well as if the spectral dependency of albedo is taken into account. 3. Page 13: figures 7 and 9 should be discussed in much more detail. 4. Figure 1 and relevant text: it seems that the proposed method works significantly better than the Kato et al. approach but it is not adequate for spectral irradiance calculations with e.g. 1nm step and resolution below 340nm. This should be highlighted in the text. 5. Figures 2-6 and relevant text: First, the meaning of counts (color scale) is not clear. The word "count" does not appear in the text. Second, the comparison of estimated vs measured irradiance is vastly dominated by the solar zenith angle, so such types of figures are always looking good. The authors are encouraged to present their results as differences (percentage , ratio, etc) between estimated and measured values vs solar zenith angle (like figure 9). The may skip some figures or replace with new ones. Moreover, due to the assumptions about the surface albedo, the differences as a function of season or some kind of snow measurement will be very helpful, since snow reports are kept at the selected sites.

Minor comments 1. Please explain abbreviations (UV, FWHM etc). In some places, the UV radiation across the whole UV spectrum is mentioned as total or total UV. Please use just UV (280-400nm) and UV-B, UV-A. The same stands for shortwave irradiance: it is referred as total, broadband etc. Please use one definition name. 2. Page 2, lines 10-15: It would be better to talk about risks and benefits from UV exposure instead of talking about "healthy" sun exposure (it is actually safe exposure). Please split and present clearly the impacts from UV over- and under-exposure (related to vitamin D deficiency). 3. Page 5, line 4: please replace "fields of cultures" with "field of agriculture" 4. Page 5, lines 11-12: please rephrase, too many "between". 5. Page 6, lines 22-25: UV irradiance, especially at lower wavelengths and under low solar zenith angles (a usual case for high latitude stations) depends FROM the ozone vertical profile, too. 6. Page 7, line 2: Insert world exponent: Angstrom exponent coefficient 7. Page 7, line 6: upwelling to downwelling flux . . . add phrase "at the surface" 8. Table 1:

Brewer instruments are mentioned as spectrophotometers and SUV-1000 instruments as sperctroradiometers. Is there such a difference? 9. Table 2 and relevant text: please add some more details about the model runs. What is the number of streams used? What about the Delta-Eddington approximation?

―――――――――――――

---

## Author Comment (AC1) · 3 Nov 2017

First of all, we thank Referee #3 for these positive remarks on this topic. The authors believe that they have understood the concerns of the referee. Their remarks have been taken into account for revising a part of the text following recommendations of the referee.

Comment 1. Page1, lines 29-31: It is claimed that improvements in the modelling of the Earth's reflectivity in the UV region are necessary. However, I cannot see that the manuscript identifies high reflectivity to be a problem when discussing the model versus measurement results in Figs.2-6 and 8 or elsewhere in the manuscript. Some weak hints are given, but no strong evidence supports the claim in the abstract. It should

be possible to identify measurements made during high and low albedo situations and compare these with the model to quantify differences caused by differences in the Earth's reflectivity.

Answer: Thank you for this valuable remark. We fully agree with you. It is also a comment from the referee #1, hence the answer is the same. We have done more investigations to explain these underestimations. We have found that it is related to albedo values. We have added a paragraph in the text to better clarify these observed underestimations in the Figure at the second paragraph of section 4.1 as follows: "Even if the points follows quite well the perfect line (Figure 2a), a set of points is seen where the method underestimate noticeably by more than 20%. These underestimations occurs between ending May and mid-July. During that period, the shortwave albedo was less than the effective UV albedo by a factor 0.8. The effective UV albedo is part of the Version 2 dataset and was derived by comparing measured clear sky spectra with corresponding radiative transfer model results (Bernhard et al., 2007). As a smaller albedo means a smaller contribution to the diffuse part of the irradiance, the difference between the shortwave and effective UV albedo may explain these underestimations seen in Figure 2a"

Comment 2. Page 3, line 25: "A very large underestimation" is mentioned. To make the manuscript complete and self-contained: may you please include numbers quantifying this underestimate?

Answer: Thank you for this remark. We have included numbers quantifying this underestimation. We have written the sentence as follows: "On the contrary, a very large underestimation of the transmissivity was observed in KB3 [283, 307] nm and KB4 [307, 328] nm by respectively 93% and 16% in relative value and exhibits relative root mean square error of 123% and 17% in clear-sky conditions. Similar relative errors are observed for cloudy conditions."

Comment 3. Page 3, lines 29-30: Please include numbers quantifying how much the

"novel parameterization" improved the transmissivity.

Answer: Thank you for this remark. We have included numbers quantifying this under-estimation. We have added the sentence as follows: "The novel parameterization of the transmissivity using more quadrature points yields maximum error of respectively 0.0006 and 0.0143 for intervals KB3 and KB4."

Comment 4. Page 6, lines 4-5: If I understand this correctly, the SHICrivm software is used to obtain the full UV spectrum from the measurements. The Sodankylä Brewer stops at 325 nm while the Jokionen Brewer stops at 365 nm. Thus, for the Sodankylä Brewer nearly the whole UV-A is extrapolated from the UV-B using SHI-Crivm. This approach should be justified and the errors in the extrapolated part assessed and doc-umented. For the Jokionen Brewer the extrapolation is less severe, but needs never-theless to be justified and the uncertainty discussed.

Answer: Thank you for this remark. We fully agree with you on the importance of inclusion of a discussion on the uncertainty introduced by the UVA extension of the measured spectra. We may estimate the approximate uncertainty at least for the UVA doses as follows. Mäkelä et al. (2016) ended up with uncertainties as high as ap-prox. 2% caused by the constant scaled UVA extension in non-weighted UVA doses. The method investigated therein is part of the routine processing scheme used in han-dling the UV irradiance spectra measured with the Brewer spectroradiometers of the FMI. The method used by ShicRIVM to extend the spectrum beyond the upper limit of the measured wavelength range uses the same kind of scaling. ShicRIVM also involves use of an atmospheric transmission model that takes into account the diur-nal/seasonal/climatological variations in the shape of the spectrum. From the spectra measured during an intercomparison in 2000, the UVA for the spectra that were cut off (and extrapolated using ShicRIVM) were compared with the full spectral analysis with following results: The daily average UVA ratio of the extrapolated 325 to 400 nm was: 1.028, where for individual spectra the relative standard deviation was 6.2%. For the 365 cut off extrapolation the ratio was 1.014 with the relative standard deviation 1.4%

(Slaper, 2017, personal communication). The time window was on daily scale. Since we are dealing with instantaneous measurements, the uncertainties are estimated to be somewhat higher. We have re-written this part of the text in the manuscript accordingly.

Comment 5. Page 6, line 8: Please do not use tinyurl and elsewhere. It is just obfuscating.

Answer: Thank you for this remark. We fully agree with you. Done as requested.

Comment 6. Page 6, lines15-16: The sentence "If there is no cloud, the sky should be clear and steady for a long period" is pretty obvious and may be omitted. But maybe you intended to say something else?

Answer: Thank you for this remark. We have re-written this part of the text in the manuscript.

Comment 7. Page 7, line 5: Please be a little more specific than "one of several available databases" and mention which one you used, including references and/or URLs.

Answer: Thank you for this remark. We fully agree with you. We have re-written the sentence to make it clear and add the url as follows: "Ground elevation is extracted from SRTM (Shuttle Radar Topography Mission) database and has been downloaded from the website http://srtm.csi.cgiar.org/SELECTION/inputCoord.asp"

Comment 8. Page 7, lines 20-25: You seem to be concerned that the albedo changes when approaching the NIR. But that should be of no relevance for the work presented here as you only discuss UV-B and UV-A. Hence, the discussion about NIR albedo may be omitted.

Answer: Thank you for this valuable remark. It is also a comment from the reviewer #2, hence the answer is the same. We have re-written this part of the text to make it clearer as follows: "As a first approximation, the UV albedo is assumed to be spectrally

constant and equal to the shortwave albedo. This assumption may result in biases depending on the surface. For example, in the case of snow surface, Varotsos et al. (2014) reported from many aircraft measurements that spectral albedo exhibits a tendency to decrease with increasing wavelength, about 0.7 from UV to about 0.4 in the NIR independently of the sky conditions. Therefore, the albedo integrated over the spectrum, becomes less than 0.7 resulting in underestimation in UV albedo, hence in a lesser contribution to diffuse UV irradiance and therefore to underestimation of the global UV."

Comment 9. Page 9, line 22: Nine variables are used to build the atmospheric states. One is the solar zenith angle which is sampled uniformly between 0 and 89. It should be noted that the solar zenith angle dependence of the radiative transfer equation is best described by the cosine of the solar zenith angle and not the solar zenith angle. Hence, in your table 2 it would have been preferable to have $\cos(\theta s)$ instead of the $\theta s$.

Answer: Thank you for this remark. Done as requested.

Comment 10. Page 10, line 8: What is meant by "excepted for KB5"?.

Answer: Thank you very much for this remark. It was a typing error. We have corrected.

Comment 11. Page 13, lines 22-23: I do not comprehend the sentence "Further investigation reveals a systematic overestimation at the low irradiance from the method itself". What is this systematic overestimation of the method itself? Is there a problem with the method? Why has not that problem been corrected?.

Answer: Thank you for this remark. We fully agree with you. We have re-written this part of the text as follows: "For the wavelength lower than 320 nm, in Figure 1, the proposed method seems to mostly overestimate when compared to the detailed spectral calculations serving as reference. This observation induces a systematic overestimation at the low irradiance from the method."

Comment 12. Page 22, Table 2: The solar zenith angle is sampled uniformly between

0 and 89. Was your radiative transfer calculations done in plane-parallel or pseudo-spherical geometry? Please include this information in the model description part.

Answer: Thank you for this remark. We fully agree with you. The radiative transfer calculations was done in plane-parallel geometry. We have included this information in the first paragraph of the section 3.

Comment 13. Page 22, Table 2: Include a column that for each variable gives the total number of samples for each variable (for Aerosol type that is obviously 7, for many of the others it is not possible to tell from the table as is). Also, where applicable, include steps. That is, for uniform distributions you include start and stop, but should also include step size.

Answer: Thank you for this remark. A clear-sky atmosphere is a combination of variables. Therefore, the number of clear-sky atmosphere is the number of samples of each variable. The number of samples depends on what we need to do with. Every time in the text, we have mentioned the number of clear-sky atmosphere which is also equal to the number of samples used for each variable.

Comment 14. Page24, Table 4: Why is the rBias so much worse for the direct than the global irradiance? Is this due to a worse sampling as in Fig.1 for the direct irradiance? The global irradiance includes the direct irradiance. Is thus the error in the global irradiance mostly due to the error in the direct irradiance?

Answer: Thank you for this remark. The errors observed in UV-B direct irradiance are due to the sampling. By increasing the number of NBk precisely in KTKB3 due to the strong ozone absorption in this band, the resampling technique may provide better results. In the proposed method, we have selected a single NBk over the band for the linear interpolation. As result, that produces an overestimation on this part of the UV spectrum. In addition, the UV direct irradiance intensities are extremely low and may produce high relative values.

Comment 15. Pages 25-26, Tables 5-6: Please include the number of data points included in the analysis for each station. This is valuable information to be able to better assess the numbers in the tables, as a station with more data points maybe considered more "valuable" than one with fewer.

Answer: Thank you for this remark. Done as requested.

Comment 16. Page 27, Fig. 1: Please indicate (label) where the various KTKB3, KTKB4, KTKB5, and KTKB6 bands are on the green line

Answer: Thank you for this remark. Done as requested.

Comment 17. Pages 27-32, Figs. 2-6: Please combine these Figures into one as you have already done in Figure 8. Figure 8 is much easier to read and allows for much easier comparison of results from the different stations than Figs.2-6.

Answer: Thank you for this remark. Done as requested.

---

## Author Response (AR1)

ANSWERS TO REFEREE #1

First of all, we thank Referee #1 for these positive remarks on this topic. The authors believe that they have understood the concerns of the referee. Their remarks have been taken into account for revising a part of the text following recommendations of the referee.

*Comment 1. In the introduction, few lines of text regarding the reasons for which the authors choose to perform the evaluation of the method using ground based measurements only from high latitude stations(and not from mid-latitudes or the tropics) would be useful.*

Thank you for this remark. We fully agree with you. We have re-written the last part of the paragraph of the introduction as follows:
"Stations have been selected to fulfill two main constraints as follows: (1) the measurement has to be carried out at cloud-free instant meaning that either it should be clearly marked or using algorithm for selecting cloud-free instants which most of the time needs broadband measurements as inputs, and (2) high quality control and assurance on the measurement. Five stations were such selected which are located at high latitudes."

*Comment 2. I suggest using abbreviations for phrases that are often referred in the manuscript (e.g. total column content of ozone could be written as TOC):*
Thank you for this suggestion. We have used much more abbreviations in the manuscript.

*Comment 3. I suggest including Figures 2-6 in a single figure, similar to figure 8.*
Thank you for this suggestion. We have included Figures 2–6 in a single figure.

*Comment 4. In Figures 2-6 the distribution of the data points around the y=x line is uneven. You claim that part of this uneven distribution is explained by the imperfect description of the effective UV albedo in the model. In some of the graphs (e.g. in figure2) there seems to be a "branch" of data where the UV-A is importantly underestimated (20% or more) by the model, even for high values of the UV-A (which possibly do not correspond to low SZAs). This branch becomes clearer in the case of UV-B (in all graphs of figure 8). Could you be more specific on what is the cause of this branch (explain more accurately what its cause is, or even provide a graph which proves that this branch is for high/low values of a specific parameter)?*

Thank you very much for this remark. We fully agree with you. We have done more investigations to explain these underestimations. We have found that these underestimations strongly depend on albedo values. We have added a paragraph in the text to better clarify these underestimations at the second paragraph of section 4.1 as follows:

"Even if the points follows quite well the perfect line (Figure 2a), a set of points is seen where the method underestimate noticeably by more than 20%. These underestimations occurs between ending May and mid-July. During that period, the shortwave albedo was less than the effective UV albedo by a factor 0.8. The effective UV albedo is part of the Version 2 dataset and was derived by comparing measured clear-sky spectra with corresponding radiative transfer model results (Bernhard et al., 2007). As a smaller albedo means a smaller contribution to the diffuse part of the irradiance, the difference between the shortwave and effective UV albedo may explain these underestimations seen in Figure 2a."

ANSWERS TO REFEREE #2

First of all, we thank Referee #2 for the positive remarks on this article. The authors believe that they have understood the concerns of the referee. The remarks have been taken into account for revising a part of the text following recommendations of the referee.

Major comments

*Comment 1. Page 6, lines 9-16: the proposed method (Lefevre et al., 2013) for the definition of clear skies is applied on broadband or total irradiance. Is this valid for UV radiation as well? UV radiation is affected considerably more by scattered cloudiness. In this case, you may have an unobstructed Sun (no clouds to cover) and a non-significant effect on diffuse broadband irradiance, so, you can assume that you have a cloud free instant. In UV (direct and diffuse irradiances) however, the effect of scattered cloudiness will be more evident. This is one of the cases that the cloud modification factors in UV and broadband irradiance are not related with a linear fit. Can you provide some evidence that the propose method is valid for UV as well?*

Thank you very much for this remark. We fully agree with you. The proposed method of Lefevre et al., (2013) for selecting clear-sky instants uses broadband irradiance. Since we have these kind of measurements for both Finnish stations, we are able to apply the Lefevre et al.

(2013) method. We have assumed that if a clear-sky instant detected with broadband irradiance, is also clear-sky instant for any spectral measurements. We have re-written a part of the paragraph as follows:

*"We assume that a cloud-free instant detected by analyzing broadband irradiances is also cloud-free for the spectral measurements. It is possible that UV is affected by the presence of scattered cloudiness which may go unnoticed in the broadband range and that the retained series of cloud-free instants for broadband may comprise cloudy instants for UV. Given the high selectivity of the algorithm of Lefèvre et al. (2013), we believe that such cases are rare and that the conclusions will be unaffected as a whole."*

*Comment 2. Page 7, lines 15-25: it is not clear in the document the type of albedo used as well as if the spectral dependency of albedo is taken into account.*

Thank you for this remark. We fully agree with you. We have re-written this part of the text to make it clearer as follows:
"As a first approximation, the UV albedo is assumed to be spectrally constant and equal to the shortwave albedo. This assumption may result in biases depending on the surface. For example, in the case of snow surface, Varotsos et al. (2014) reported from many aircraft measurements that spectral albedo exhibits a tendency to decrease with increasing wavelength, about 0.7 from UV to about 0.4 in the NIR independently of the sky conditions. Therefore, the albedo integrated over the spectrum, becomes less than 0.7 resulting in underestimation in UV albedo, hence in a lesser contribution to diffuse UV irradiance and therefore to underestimation of the global UV."

*Comment 3. Page13: figures 7 and 9 should be discussed in much more detail.*
Thank for this remark. We fully agree with you. We have provided new plots and re-written the relevant part of the text.

*Comment 4. Figure 1 and relevant text: it seems that the proposed method works significantly better than the Kato et al. approach but it is not adequate for spectral irradiance calculations with e.g. 1 nm step and resolution below 340 nm. This should be highlighted in the text.*

Thank you for this remark. We fully agree with you. We have highlighted it in there-written this part of the text as follows:

"For the wavelength lower than 320 nm, in Figure 1, the proposed method seems to mostly overestimate when compared to the details spectral calculations serving as reference. This observation induces a systematic overestimation at the low irradiance from the method."

*Comment 5. Figures 2-6 and relevant text: First, the meaning of counts (colorscale) is not clear. The word "count" does not appear in the text. Second, the comparison of estimated vs measured irradiance is vastly dominated by the solar zenith angle, so such types of figures are always looking good. The authors are encouraged to present their results as differences (percentage, ratio, etc) between estimated and measured values vs solar zenith angle (likefigure9). The may skip some figures or replace with new ones. Moreover, due to the assumptions about the surface albedo, the differences as a function of season or some kind of snow measurement will be very helpful, since snow reports are kept at the selected sites.*

Thank you for this remark. We fully agree with you. For the first part of the comment, we have changed the caption of the Figure as well as the relevant text as follows:

"Scatter density plot between measurements of UV–A and estimates for each station with each station name at top. The colorbar indicates the number of points in the area within the interval 0.4 W m$^{-2}$ x 0.4 W m$^{-2}$"

Then, for the second part, we have replaced the plots by the new ones. They are the ratio and difference. We have re-written the relevant part of the text.

Minor comments:

*Comment 1. Please explain abbreviations (UV, FWHM etc). In some places, the UV radiation across the whole UV spectrum is mentioned as total or total UV. Please use just UV (280-400 nm) and UV-B, UV-A. The same stands for shortwave irradiance: it is referred as total, broadband etc. Please use one definition name*

Thank you for this remark. Done as requested.

*Comment 2. Page 2, lines 10-15: It would be better to talk about risks and benefits from UV exposure instead of talking about "healthy" sun exposure (it is actually safe exposure). Please split and present clearly the impacts from UV over-and under-exposure (related to vitamin D deficiency).*

Thank you for this remark. We fully agree with you. We have replaced the word healthy by "safe". Then we have clearly presented the impacts from UV over-and under-exposure in the second sentence of the first paragraph of the introduction as follows:

"For instance, UV radiation is a principal source of vitamin-D, while the excess UV exposure is a risk factor for skin cancers, cataracts and immunosuppression"

*Comment 3. Page 5, line 4: please replace "fields of cultures" with "field of agriculture"*
Thank you for this remark. Done as requested.

*Comment 4. Page 5, lines 11-12: please rephrase, too many "between"*
We fully agree with this remark. We rephrased. The sentence is now as follows:
"the effective UV albedo varies between 0.6 and 0.95 occurring from August until November"

*Comment 5. Page 6, lines 22-25: UV irradiance, especially at lower wavelengths and under low solar zenith angles (a usual case for high latitude stations) depends FROM the ozone vertical profile, too.*
Thank you for this remark. We fully agree with you and we have added this dependence in the text.

*Comment 6. Page 7, line 2: Insert world exponent: Angstrom exponent coefficient.*
Thank you for this remark. Done as requested.

*Comment 7. Page7, line 6: upwelling to downwelling flux::: add phrase "at the surface"*
Thank you for this remark. Done as requested.

*Comment 8. Table1: Brewer instruments are mentioned as spectrophotometers and SUV-1000 instruments as sperctroradiometers. Is there such a difference?*
The Brewer is a spectroradiometer. Its name given by the manufacturer, however, is Brewer spectrophotometer (http://www.kippzonen.com/Product/50/Brewer-MkIII-Spectrophotometer#.WfBIhXZLeyp). We fully agree with you. We have changed the text accordingly in the Table 1.

*Comment 9. Table 2 and relevant text: please add some more details about the model runs. What is the number of streams used? What about the Delta-Eddington approximation?*

Thank you for this remark. We provided more details in the text. We added one sentence at the first paragraph in the section 3 as follows:

"For all the radiative transfer simulations, a plane-parallel atmosphere was assumed and the DISORT 2.0 (discrete ordinate technique) algorithm (Stamnes et al., 2000) with 16 streams was selected to solve the radiative transfer equation because several articles have demonstrated the accuracy of its results when compared to robust and more time consuming solvers."

ANSWERS TO REFEREE #3

First of all, we thank Referee #3 for these positive remarks on this topic. The authors believe that they have understood the concerns of the referee. Their remarks have been taken into account for revising a part of the text following recommendations of the referee.

*Comment 1. Page1, lines 29-31: It is claimed that improvements in the modelling of the Earth's reflectivity in the UV region are necessary. However, I cannot see that the manuscript identifies high reflectivity to be a problem when discussing the model versus measurement results in Figs.2-6 and 8 or elsewhere in the manuscript. Some weak hints are given, but no strong evidence supports the claim in the abstract. It should be possible to identify measurements made during high and low albedo situations and compare these with the model to quantify differences caused by differences in the Earth's reflectivity.*

Thank you for this valuable remark. We fully agree with you. It is also a comment from the referee #1, hence the answer is the same.
We have done more investigations to explain these underestimations. We have found that it is related to albedo values. We have added a paragraph in the text to better clarify these observed underestimations in the Figure at the second paragraph of section 4.1 as follows:

"Even if the points follows quite well the perfect line (Figure 2a), a set of points is seen where the method underestimate noticeably by more than 20%. These underestimations occurs between ending May and mid-July. During that period, the shortwave albedo was less than the effective UV albedo by a factor 0.8. The effective UV albedo is part of the Version 2 dataset and was derived by comparing measured clear-sky spectra with corresponding radiative transfer model results (Bernhard et al., 2007). As a smaller albedo means a smaller contribution to the

diffuse part of the irradiance, the difference between the shortwave and effective UV albedo may explain these underestimations seen in Figure 2a"

*Comment 2. Page 3, line 25: "A very large underestimation" is mentioned. To make the manuscript complete and self-contained: may you please include numbers quantifying this underestimate?*

Thank you for this remark. We have included numbers quantifying this underestimation. We have written the sentence as follows:

"On the contrary, a very large underestimation of the transmissivity was observed in $KB_3$ [283, 307] nm and $KB_4$ [307, 328] nm by respectively -93% and -16% in relative value and exhibits relative root mean square error of 123% and 17% in clear-sky conditions. Similar relative errors are observed for cloudy conditions."

*Comment 2. Page 3, lines 29-30: Please include numbers quantifying how much the "novel parameterization" improved the transmissivity.*

Thank you for this remark. We have included numbers quantifying this underestimation. We have added the sentence as follows:

"The novel parameterization of the transmissivity using more quadrature points yields maximum error of respectively 0.0006 and 0.0143 for intervals $KB_3$ and $KB_4$."

*Comment 4. Page 6, lines 4-5: If I understand this correctly, the SHICrivm software is used to obtain the full UV spectrum from the measurements. The Sodankylä Brewer stops at 325 nm while the Jokionen Brewer stops at 365 nm. Thus, for the Sodankylä Brewer nearly the whole UV-A is extrapolated from the UV-B using SHI-Crivm. This approach should be justified and the errors in the extrapolated part assessed and documented. For the Jokionen Brewer the extrapolation is less severe, but needs nevertheless to be justified and the uncertainty discussed.*

Thank you for this remark. We fully agree with you on the importance of inclusion of a discussion on the uncertainty introduced by the UVA extension of the measured spectra. We may estimate the approximate uncertainty at least for the UVA doses as follows. Mäkelä et al. (2016) ended up with uncertainties as high as approx. 2% caused by the constant scaled UVA extension in non-weighted UVA doses. The method investigated therein is part of the routine

processing scheme used in handling the UV irradiance spectra measured with the Brewer spectroradiometers of the FMI. The method used by ShicRIVM to extend the spectrum beyond the upper limit of the measured wavelength range uses the same kind of scaling. ShicRIVM also involves use of an atmospheric transmission model that takes into account the diurnal/seasonal/climatological variations in the shape of the spectrum. From the spectra measured during an intercomparison in 2000, the UVA for the spectra that were cut off (and extrapolated using ShicRIVM) were compared with the full spectral analysis with following results: The daily average UVA ratio of the extrapolated 325 to 400 nm was: 1.028, where for individual spectra the relative standard deviation was 6.2%. For the 365 cut off extrapolation the ratio was 1.014 with the relative standard deviation 1.4% (Slaper, 2017, personal communication). The time window was on daily scale. Since we are dealing with instantaneous measurements, the uncertainties are estimated to be somewhat higher. We have re-written this part of the text in the manuscript accordingly.

*Comment 5. Page 6, line 8: Please do not use tinyurl and elsewhere. It is just obfuscating.*

Thank you for this remark. We fully agree with you. Done as requested.

*Comment 6. Page 6, lines15-16: The sentence "If there is no cloud, the sky should be clear and steady for a long period" is pretty obvious and may be omitted. But maybe you intended to say something else?*

Thank you for this remark. We have removed the sentence and add a new one sentence to make it simple as follows:
"Therefore, we assume that a cloud-free instant detected by using broadband irradiances measurements should be also cloud-free one for the spectral measurements."

*Comment 7. Page 7, line5: Please be a little more specific than "one of several available databases" and mention which one you used, including references and/or URLs.*

Thank you for this remark. We fully agree with you. We have re-written the sentence to make it clear and add the url as follows:
"Ground elevation is extracted from SRTM (Shuttle Radar Topography Mission) database and has been downloaded from the website http://srtm.csi.cgiar.org/SELECTION/inputCoord.asp"

*Comment 8. Page 7, lines 20-25: You seem to be concerned that the albedo changes when approaching the NIR. But that should be of no relevance for the work presented here as you only discuss UV-B and UV-A. Hence, the discussion about NIR albedo may be omitted.*

Thank you for this valuable remark. It is also a comment from the reviewer #2, hence the answer is the same. We have re-written this part of the text to make it clearer as follows:
"As a first approximation, the UV albedo is assumed to be spectrally constant and equal to the shortwave albedo. This assumption may result in biases depending on the surface. For example, in the case of snow surface, Varotsos et al. (2014) reported from many aircraft measurements that spectral albedo exhibits a tendency to decrease with increasing wavelength, about 0.7 from UV to about 0.4 in the NIR independently of the sky conditions. Therefore, the albedo integrated over the spectrum, becomes less than 0.7 resulting in underestimation in UV albedo, hence in a lesser contribution to diffuse UV irradiance and therefore to underestimation of the global UV."

*Comment 9. Page 9, line 22: Nine variables are used to build the atmospheric states. One is the solar zenith angle which is sampled uniformly between 0 and 89. It should be noted that the solar zenith angle dependence of the radiative transfer equation is best described by the cosine of the solar zenith angle and not the solar zenith angle. Hence, in your table 2 it would have been preferable to have $\cos\theta_s$ instead of the $\theta_s$.*

Thank you for this remark. Done as requested.

*Comment 10. Page 10, line 8: What is meant by "excepted for KB5"?.*

Thank you very much for this remark. It was a typing errors. We have corrected.

*Comment 11. Page 13, lines 22-23: I do not comprehend the sentence "Further investigation reveals a systematic overestimation at the low irradiance from the method itself". What is this systematic overestimation of the method itself? Is there a problem with the method? Why has not that problem been corrected?.*

Thank you for this remark. We fully agree with you. We have re-written this part of the text as follows:

"For the wavelength lower than 320 nm, in Figure 1, the proposed method seems to mostly overestimate when compared to the details spectral calculations serving as reference. This observation induces a systematic overestimation at the low irradiance from the method."

*Comment 12. Page 22, Table 2: The solar zenith angle is sampled uniformly between 0 and 89. Was your radiative transfer calculations done in plane-parallel or pseudo-spherical geometry? Please include this information in the model description part.*

Thank you for this remark. We fully agree with you. We have included this information in the first paragraph of the section 3.

*Comment 13. Page 22, Table 2: Include a column that for each variable gives the total number of samples for each variable (for Aerosol type that is obviously 7, for many of the others it is not possible to tell from the table as is). Also, where applicable, include steps. That is, for uniform distributions you include start and stop, but should also include step size.*

Thank you for this remark. A clear-sky atmosphere is a combination of variables. Therefore, the number of clear-sky atmosphere is number of samples of each variable.
The number of samples depends on what we need to do with. Every time in the text, we have mentioned the number of clear-sky atmospheres which is also equal to the number of samples used for each variable.

*Comment 14. Page24, Table 4: Why is the rBias so much worse for the direct than the global irradiance? Is this due to a worse sampling as in Fig.1 for the direct irradiance? The global irradiance includes the direct irradiance. Is thus the error in the global irradiance mostly due to the error in the direct irradiance?*

Thank you for this remark. The errors observed in UV-B direct irradiance are due to the sampling. By increasing the number of $NB_k$ precisely in $KT_{KB3}$ due to the strong ozone absorption in this band, the resampling technique may provide better results. In the proposed method, we have selected a single $NB_k$ over the band in the linear interpolation. As result, that

produces an overestimation on this part of the UV spectrum. In addition, the UV direct irradiance intensities are extremely low and may produce high relative values.

*Comment 15. Pages 25-26, Tables 5-6: Please include the number of data points included in the analysis for each station. This is valuable information to be able to better assess the numbers in the tables, as a station with more data points maybe considered more "valuable" than one with fewer.*

Thank you for this remark. Done as requested.

*Comment 16. Page27, Fig. 1: Please indicate (label) where the various $KT_{KB3}$, $KT_{KB4}$, $KT_{KB5}$, and $KT_{KB6}$ bands are on the green line*

Thank you for this remark. Done as requested.

*Comment 17. Pages 27-32, Figs. 2-6: Please combine these Figures into one as you have already done in Figure 8. Figure 8 is much easier to read and allows for much easier comparison of results from the different stations than Figs.2-6.*

Thank you for this remark. Done as requested.